# Facile Synthesis of Zn-Co-S Nanostrip Cluster Arrays on Ni Foam for High-Performance Hybrid Supercapacitors

**DOI:** 10.3390/nano11123209

**Published:** 2021-11-26

**Authors:** Subbukalai Vijayakumar, Ganesh Dhakal, Soo-Hyun Kim, Jintae Lee, Yong Rok Lee, Jae-Jin Shim

**Affiliations:** 1School of Chemical Engineering, Yeungnam University, Gyeongsan 38541, Gyeongbuk, Korea; svijaygri@gmail.com (S.V.); gdhakal17@gmail.com (G.D.); jtlee@yu.ac.kr (J.L.); yrlee@yu.ac.kr (Y.R.L.); 2Centre for Research and Post-Graduate Studies in Physics, Ayya Nadar Janaki Ammal College, Sivakasi 626124, India; 3School of Materials Science and Engineering, Yeungnam University, Gyeongsan 38541, Gyeongbuk, Korea; soohyun@ynu.ac.kr

**Keywords:** zinc cobalt sulfide, zinc cobalt oxide, hybrid supercapacitor, nanostrip cluster array, nickel foam

## Abstract

Mixed metal sulfides exhibit outstanding electrochemical performance compared to single metal sulfides and mixed metal oxides because of their richer redox reactions and high electronic conductivity. In the present study, Zn-Co-S nanostrip cluster arrays were formed from ZnCo_2_O_4_ grown on Ni foam by an anion exchange reaction using a two-step hydrothermal process. Morphological characterization confirmed that the Zn-Co-S nanostrip cluster arrays had grown homogeneously on the skeleton of the 3D Ni foam. The length of the nanostrip was approximately 8 µm, and the width ranged from 600 to 800 nm. The Ni foam-supported Zn-Co-S nanostrip cluster arrays were assessed directly for electrochemical supercapacitor applications. Compared to ZnCo_2_O_4_, the Zn-Co-S electrode exhibited a three-fold higher specific capacity of 830 C g^−1^ at a specific current of 2.0 A g^−1^. The higher polarizability, lower electro-negativity, and larger size of the S^2−^ ion played an important role in substituting oxygen with sulfur, which enhanced the performance. The Zn-Co-S//AC hybrid device delivered a maximum specific energy of 19.0 Wh kg^−1^ at a specific power of 514 W kg^−1^. The remarkable performance of Zn-Co-S nanostrip cluster arrays highlights their potential as a positive electrode for hybrid supercapacitor applications.

## 1. Introduction

An electrochemical supercapacitor is one of the most promising energy storage devices, owing to its high power density, longer cycling stability, and excellent operational safety [1,2]. Supercapacitors are used widely in portable electronic devices and hybrid electric vehicles [3], but they deliver a lower energy density than secondary batteries. Hence, more studies are needed to improve the energy density of electrochemical supercapacitor devices without any loss of power density. Hybrid supercapacitors are a new avenue for achieving high energy densities without sacrificing power density. These devices consist of a capacitor-like electrode (electrical double-layer capacitor, EDLC) as a power source and a battery-like electrode as an energy source [4,5,6]. This hybrid device can widen the operation voltage and increase the energy density.

Nickel and cobalt-based oxide/hydroxide/sulfides exhibit battery-like behavior in alkaline electrolytes [7,8]. Recently, transition metal sulfides have been studied widely as a battery-type electrode material for hybrid supercapacitor devices, because sulfides provide higher electrical conductivity and better electrochemical reactivity than oxides [8,9]. The substitution of oxygen with sulfur increases the conductivity of the electrode and ion diffusivity because of higher polarizability, lower electro-negativity, and larger size of the S^2−^ ion [10]. Mixed metal sulfides exhibit excellent electrochemical reactivity compared to single-component metal sulfide and their mixed oxide counterparts because of their richer redox reactions and high electronic conductivity [11,12].

Cobalt sulfides provide the highest capacitances in aqueous electrolytes among all the transition metal oxides and sulfides. On the other hand, Co is an expensive material. Many studies have been performed to replace Co with less expensive metals, such as Ni, Zn, and Cu. Among them, zinc is the most abundant and least expensive, and is relatively nontoxic. Therefore, Zn was chosen to reduce the use of Co, expecting a synergistic effect of Zn with a larger atomic size than Co.

Zinc cobalt sulfide is a less studied material for hybrid supercapacitor applications. Several methods, such as sequential chemical etching and sulfidation strategy [13], facile oil phase approach [14], and direct growth on conducting substrates [15,16], have been adopted for the synthesis of zinc cobalt sulfide nanostructures. Zhang et al. [13] prepared double-shelled zinc cobalt sulfide dodecahedral cages from bimetallic zeolitic imidazolate frameworks for hybrid supercapacitor applications. The double-shelled zinc cobalt sulfide delivered a maximum specific capacitance of 1270 F g^−1^ at 1 A g^−1^ with good cycling stability. Tong et al. [14] prepared Zn_x_Co_1-x_S nanoartichokes in the oil phase and reported a maximum specific capacitance of 486 F g^−1^ at a current density of 2.0 A g^−1^. Recently, Syed et al. [15] reported the preparation of Ni(OH)_2_ nanoflake-wrapped zinc cobalt sulfide nanotube arrays on Ni foam and achieved a high specific capacitance of 2160 F g^−1^.

Among these methods, electrode materials grown directly on conducting current collectors, including Ni foam, carbon cloth, and Ti foil, show enhanced electrochemical properties by avoiding the use of binder and carbon black, which hinder ion access during electrochemical reactions [17,18,19]. In this approach, ion transportation is improved by enhancing the interfacial contact between the current collector and active material [20]. This paper reports the synthesis of ZnCo_2_O_4_ nanostrips on the Ni foam and their successful conversion to Zn-Co-S by anion exchange without destroying the structure. Ni foam was used to grow electrode materials because of its three-dimensional network, high porosity, and ease of electrolyte accessibility to the electrodes [19]. The electrochemical supercapacitor performance of ZnCo_2_O_4_ and Zn_0.76_Co_0.24_S/Co_3_S_4_ nanostrip arrays was characterized by cyclic voltammetry, charge–discharge measurements, and impedance analysis.

## 2. Materials and Methods

### 2.1. Materials

Zinc nitrate hexahydrate (98.0%), polytetrafluoroethylene (PTFE) (99.9%), and activated carbon (density 1.8–2.1 g cm^−3^, particle size 100 mesh) were supplied by Sigma Aldrich. Cobalt nitrate hexahydrate (98.0%), sodium sulfide (98.0%), potassium hydroxide (85.0%, pellets), carbon black (99.9%), and Whatman filter paper (180 μm thick, pore size 11 μm) were obtained from Alfa Aesar. Ammonium fluoride (99.9%) and urea (99.0%) were procured from Junsei. Ethanol (99.9%), hydrochloric acid (35.0–37.0%), and acetone (99.9%) were acquired from Duksan Pure Chemical. Ni foam was purchased from MTI cooperation (New York, NY, USA).

### 2.2. Synthesis of Zn-Co-S Nanostrip

Zn_0.76_Co_0.24_S/Co_3_S_4_ nanostrip arrays were prepared using a two-step hydrothermal process. First, ZnCo_2_O_4_ nanostrip arrays were prepared using a hydrothermal method. The ZnCo_2_O_4_ nanostrip arrays were then converted to Zn_0.76_Co_0.24_S/Co_3_S_4_ nanostrip cluster arrays by an anion-exchange reaction using a second hydrothermal reaction. In a typical preparation, 35 mL of a clear aqueous solution containing 5 mM zinc nitrate hexahydrate, 10 mM cobalt nitrate hexahydrate, 10 mM NH_4_F, and 25 mM urea was prepared. Before deposition, Ni foam (3 × 2 cm) was cleaned with 3.0 M HCl in an ultrasound bath and rinsed with deionized water and ethanol. The reaction mixture and Ni foam were then transferred to a 50 mL Teflon-lined autoclave and kept at 120 °C for 12 h. The autoclave was then allowed to cool to room temperature. The coated Ni foam sample was washed and dried at 60 °C. Finally, the Ni foam-supported ZnCo_2_O_4_ nanostrip arrays were obtained by calcination at 350 °C for 2 h in air. The Ni foam-supported ZnCo_2_O_4_ were subjected to a sulfurization process. In this process, 0.0910 g of Na_2_S.xH_2_O was dissolved in 35 mL of water and stirred to obtain a clear solution. The ZnCo_2_O_4_-coated Ni foam and sodium sulfide solution was transferred to a 50 mL Teflon-lined autoclave and kept at 160 °C for 6 h. After the reaction was complete, the sample was collected, washed, and dried in a vacuum at 60 °C for 6 h.

### 2.3. Materials Characterization

The crystal structure was analyzed by X-ray diffraction (XRD, PANalytical X’Pert-PRO MPD). The morphology of the samples was analyzed by field emission scanning electron microscopy (FESEM, Hitachi S-4800, Tokyo, Japan). The microstructure of the sample was characterized by high-resolution transmission electron microscopy (HRTEM, FEI Tecnai G2 F20, Eindhoven, The Netherlands) at an acceleration voltage of 200 kV. X-ray photoelectron spectroscopy (XPS, Thermo Scientific, Boston, MA, USA) was performed using Al Kα radiation.

### 2.4. Electrochemical Measurements

All electrochemical characterizations were performed using a electrochemical workstation (Metrohm Autolab PGSTAT 302N, Utrecht, The Netherlands) electrochemical workstation. Three-electrode measurements were carried out using the working, counter, and reference electrodes made of Ni-foam-supported Zn_0.76_Co_0.24_S/Co_3_S_4_ or ZnCo_2_O_4_, platinum, and Ag/AgCl, respectively. All measurements were carried out using 2.0 M KOH as the electrolyte. The hybrid supercapacitor device was assembled in a split test cell (MTI Corporation) using Zn_0.76_Co_0.24_S/Co_3_S_4_ nanostrip arrays as the positive electrode, activated carbon (AC) as the negative electrode, and filter paper as the separator with a 2.0 M KOH electrolyte. The activated carbon electrode was prepared by mixing 80% activated carbon with 15% carbon black using a mortar and pestle. To this mixture, 5% PTFE was combined with a few drops of ethanol to form a paste. Ni foam was then coated with the paste and dried overnight at 60 °C.

## 3. Results and Discussion

### 3.1. Structural and Morphological Analysis

The Zn-Co-S nanostrip cluster arrays on Ni foam were prepared using a two-step hydrothermal process. Figure 1 presents a schematic illustration of the preparation process of preparation of Zn-Co-S nanostrip cluster arrays. Here, Ni foam was used as a current collector to prepare Zn-Co-S because of its zigzag skeleton and high porosity, which helps increase the active surface area. First, ZnCo_2_O_4_ nanostrip arrays (Appendix A shows the XRD pattern of ZnCo_2_O_4_) were prepared on Ni foam using a facile hydrothermal process followed by calcination. The ZnCo_2_O_4_ nanostrip arrays were converted to Zn-Co-S nanostrip cluster arrays by an anion-exchange reaction using sodium sulfide as the sulfur source. The mass loading of the active material on a Ni foam was approximately 2.5 mg cm^−2^, which was calculated from the mass difference before and after the synthesis reaction. Because the Ni foam was cleaned with HCl, the possibility of a reaction of the cleaned Ni foam with metal-oxide precursors was low, resulting in a relatively constant quality of the Ni foam.

Figure 2 shows an XRD pattern of Zn-Co-S grown on Ni foam. The XRD peaks at 28.9, 47.9, and 56.7° 2θ were assigned to the (111), (220), and (311) planes of Zn_0.76_Co_0.24_S, respectively (JCPDS No: 47-1656). Similarly, the XRD peaks at 31.5, 38.2, 50.3, and 55.5° 2θ correspond to (311), (400), (511), and (440) planes of Co_3_S_4_, (JCPDS No: 73-1703). The three additional intense peaks correspond to Ni from the Ni foam. XRD shows the formation of zinc cobalt mixed sulfide on Ni foam. The morphology and microstructure of the Zn-Co-S were examined by FE-SEM and HR-TEM. Figure 3a shows an FE-SEM image of ZnCo_2_O_4_. The image clearly shows the nanostrip arrays grown on Ni foam. Figure 3b,c shows FE-SEM images of Ni foam supported Zn-Co-S. The low magnification FE-SEM image (Figure 3b) showed that the Zn-Co-S nanostrip cluster arrays grew homogeneously on the skeleton of the 3D Ni foam.

The high magnification FE-SEM images (Figure 3c,d) clearly show the morphology and dimensions of Zn-Co-S. The Zn-Co-S nanostrip assembled and formed a nanostrip cluster array. The length of the nanostrip was approximately 8 µm, and the width ranged from 600 to 800 nm. The morphology and structure of the Zn-Co-S grown on Ni foam were characterized further by HR-TEM. Figure 4a,b shows HR-TEM images of Ni-foam- supported Zn-Co-S nanostrip cluster arrays; the thin nanostrips assembled and form a single nanostrip. This thin nanostrip can provide better pathways for efficient ion/electron transport, enhancing the electrochemical activity. Figure 4c shows highly resolved lattice fringes with an interplanar spacing of 0.16 nm, corresponding to the (311) plane of Zn_0.76_Co_0.24_S. The inset in Figure 4c reveals the SAED pattern of Zn-Co-S. The well-defined diffraction rings highlight the polycrystalline nature of the Zn-Co-S nanostrip cluster arrays. The elemental composition and elemental distribution were analyzed by EDAX and elemental mapping analysis. Figure 4d presents the EDAX data of the Zn-Co-S nanostrip cluster arrays. This shows the presence of Zn, Co, and S in Ni foam-supported Zn-Co-S. Figure 4e–h presents the distribution of Zn, Co, and S on the Zn_0.76_Co_0.24S_/Co_3_S_4_ nanostrip. The homogeneous distribution of Zn, Co, and S further confirmed the successful formation of Zn-Co-S nanostrip cluster arrays.

The elemental composition and oxidation state of ZnCo_2_O_4_ and Zn-Co-S were examined by XPS. Appendix A presents the XPS survey spectrum and deconvoluted spectra of ZnCo_2_O_4_. The survey spectrum (Appendix A) confirmed the presence of Zn, Co, and O. The Zn 2p spectrum (Appendix A) showed two peaks at 1021 eV and 1044.08 eV, corresponding to Zn 2p_3/2_ and Zn 2p_1/2_, respectively [21]. Appendix A presents the spectrum of Co 2p, which has two main peaks centered at 779.66 eV and 794.6 eV corresponding to Co 2p_3/2_ and Co 2p_1/2_, respectively [22]. The oxygen spectrum (Appendix A) was resolved into two components: two peaks centered at 529.35 eV and 531.17 eV, which were assigned to metal-oxygen bonding and many defect sites, respectively [23]. Figure 5 presents the XPS survey spectrum and deconvoluted spectra of Zn-Co-S. The survey spectrum (Figure 5a) revealed the presence of Zn, Co, and S. The Zn 2p spectrum in Figure 5b has two peaks at 1021.8 eV and 1044.8 eV, which correspond to Zn 2p_3/2_ and Zn 2p_1/2_, respectively [21]. The high-resolution Co 2p spectrum (Figure 5c) has two peaks at 781.2 eV (Co 2p_3/2_) and 796.9 eV (Co 2p_1/2_), along with two shakeup satellite peaks. The spin-orbit splitting of Co 2p_3/2_ and Co 2p_1/2_ was over 15 eV, confirming the coexistence of Co^2+^ and Co^3+^ [24]. Figure 5d presents the S 2p spectrum. The fitted peak centered at the binding energies 161.4 eV and 162.7 eV corresponds to S 2p_3/2_ and S 2p_1/2_, respectively [12]. The additional peak at 168.7 eV was assigned to surface sulfur with a high oxide state, i.e., SO_4_^2-^ [25].

### 3.2. Electrochemical Performance of Zn-Co-S

The electrochemical performance of Zn_0.76_Co_0.24_S/Co_3_S_4_ (Zn-Co-S) nanostrip cluster arrays and ZnCo_2_O_4_ nanostrip arrays grown on 3D Ni foam was evaluated in a three-electrode cell configuration using 2.0 M KOH as an electrolyte by cyclic voltammetry (CV), galvanostatic charge–discharge (GCD), and electrochemical impedance spectroscopy (EIS). Figure 6a compares the CV curve of Zn-Co-S and ZnCo_2_O_4_ electrodes. Both CV curves showed well-defined redox peaks with a large peak potential difference, confirming the battery-type behavior of the material [26,27]. The asymmetry of the anodic and cathodic redox peaks was assigned to the kinetic irreversibility of the Faradaic redox process [28]. The possible electrochemical reactions of Zn-Co-S and ZnCo_2_O_4_ in an alkaline solution are as follows [15,29]:(1)$$CoS+OH^{-}⇌CoSOH+e^{-}$$
(2)$$CoSOH+OH^{-}⇌CoSO+H_{2}O+e^{-}$$
(3)$$ZnS+OH^{-}⇌ZnSOH+e^{-}$$
(4)$$Co_{2}O_{4}^{2-}+2H_{2}O+e^{-}⇌2CoOOH+2OH^{-}$$
(5)$$CoOOH+H_{2}O+e^{-}⇌Co{\left(OH\right)}_{2}+OH^{-}$$

The area under the CV curve of Zn-Co-S was larger than that of ZnCo_2_O_4_, suggesting that the Zn-Co-S electrode exhibits better electrochemical charge storage than ZnCo_2_O_4_. The CV measurement was carried out at a scan rate of 5, 10, 25, and 50 mV s^−1^ between the potential window of 0 and 0.55 V. Figure 6b and Appendix A present the CV curves of Zn-Co-S and ZnCo_2_O_4_ electrodes at different scan rates. The anodic and cathodic peaks shifted towards a more positive and negative potential with increasing scan rates due to Ohmic resistance and electrical polarization of the electrode material [30]. Figure 6c depicts the cathodic peak current vs. square root of the scan rate curve of Zn-Co-S and ZnCo_2_O_4_ electrodes. The linearity of the curve confirms that the electrochemical reaction is diffusion-controlled. This suggests that the Zn-Co-S and ZnCo_2_O_4_ electrodes exhibit battery-type behavior in the KOH electrolyte.

Galvanostatic charge–discharge measurements of the Zn-Co-S and ZnCo_2_O_4_ electrodes were conducted between the potential window of 0 and 0.45 V at different specific currents. Figure 6d shows the charge–discharge curve of the Zn-Co-S and ZnCo_2_O_4_ electrodes at a specific current of 2.0 A g^−1^. The voltage plateaus of the charge–discharge curve confirm that charge storage is due mainly to the Faradaic behavior of the battery-type electrodes [26]. Figure 6e and Appendix A show the charge–discharge curve of Zn-Co-S and ZnCo_2_O_4_ electrodes at different specific currents. Because both electrodes exhibit battery-type behavior, the specific capacity, *C_S_* (C g^−1^), was calculated from the integrated area of the discharge curve using the following equation [31]:(6)$$C_{s}=\frac{2I\int V\left(t\right)dt}{mV}$$
where *I* is the applied current; *m* is the mass of the electroactive material in mg; *V* is the potential window.

The specific capacities of the Zn-Co-S and ZnCo_2_O_4_ electrodes were 830 C g^−1^ (1840 F g^−1^) and 300 C g^−1^ (667 F g^−1^) at a specific current of 2.0 A g^−1^. Figure 6f shows the specific capacity vs. specific current plot. The specific capacity decreased with increasing specific current, probably due to the increasing voltage drop and sluggish kinetics of the redox reaction [32]. The Zn-Co-S electrode showed a higher specific capacity than the ZnCo_2_O_4_ electrode. This was attributed to the higher conductivity, higher redox potential, and the synergic effects of the different metal ions of the Zn-Co-S electrode.

The long-term cycling stability of the Zn-Co-S was tested for 2000 cycles at a high specific current of 20 A g^−1^. Appendix A presents the cycling stability of the Zn-Co-S electrode. The Zn-Co-S nanostrip cluster arrays exhibited approximately 76% specific capacity retention after 2000 continuous cycles. Electrochemical impedance analysis was carried out over the frequency range of 0.1 to 100 kHz. Appendix A shows the impedance plot of Zn-Co-S than ZnCo_2_O_4_ electrodes. In the impedance plot, the intercept on the real axis at high frequency is the solution resistance (Rs). The solution resistances of Zn-Co-S and ZnCo_2_O_4_ electrodes were 0.92 and 1.16 Ω, respectively. The semi-circle at the high-frequency region is related to the charge transfer resistance (Rct). The Rct value of the Zn-Co-S electrode was 0.15 Ω. The straight line at the low-frequency region corresponds to the Warburg resistance.

The high performance of the Zn-Co-S nanostrip cluster arrays was due to the following reasons. First, the direct growth of Zn-Co-S nanostrip cluster arrays enhances ion transport due to the high interfacial contact between the electroactive material and substrate. Second, the Zn-Co-S nanostrip cluster array grown directly on Ni foam avoids the use of binders and carbon black, which enhances the effective use of active material. Third, the lower electro-negativity of sulfur can increase the electron mobility, enhancing the electrical conductivity [33]. Finally, the nanostrip cluster arrays can provide better pathways for efficient ion/electron transport that enhance the electrochemical activity.

### 3.3. Electrochemical Performance of Zn-Co-S//AC Device

The supercapacitor performance of the Zn-Co-S nanostrip cluster arrays was evaluated by assembling a hybrid device using Zn-Co-S nanostrip cluster arrays as a positive electrode and activated carbon (AC) as a negative electrode. For good electrochemical performance, the charge balance between the two electrodes should follow the equation, *q^+^* = *q^−^*, where q^+^ and q^−^ are the charges stored by the positive and negative electrodes. To achieve charge balance, the mass ratio of both positive and negative electrodes was calculated based on the following equation [34,35]:(7)$$\frac{m_{+}}{m_{-}}=\frac{C_{-}\Delta V_{-}}{Q_{+}}$$
where *Q_+_* and *C*_−_ are the charge of the positive and specific capacitance of the negative electrodes; Δ*V*_−_ is the potential window of the negative electrode; m_+_ and m_−_ are the mass of the electrode materials. Figure 7a shows the CV curves of activated carbon and Zn-Co-S electrodes measured at 5 mV s^−1^ in a half-cell system. The rectangular-shaped CV curve of the AC electrode exhibited electric double-layer capacitor behavior with a potential window from −1.0 to 0 V. The Zn-Co-S electrode showed battery-type Faradaic redox peaks with a potential window from 0 to 0.55 V. The CV measurements were carried out at different voltages (1.0 to 1.5 V) to obtain the optimal voltage window (Figure 7b). The curve at 1.55 V showed a small spike near the highest voltage, which was not included in this figure. As the curve at 1.5 V showed the largest area with distinct redox behavior, 1.5 V was chosen as the voltage window for the device. The CV curves of the asymmetric hybrid supercapacitor device (Zn-Co-S//AC) measured at different scan rates (Figure 7c) clearly show the characteristics of both the double layer capacitor and battery type electrode behavior (Figure 7b). No significant change or shift of the redox peaks was observed when the scan rate increased from 5 to 50 mV s^−1^, which indicates a good charge delivery rate of the fabricated device.

Figure 7d presents the charge–discharge curve of the asymmetric hybrid supercapacitor device at different voltages, ranging from 1.0 to 1.5 V, which is consistent with the CV curves in Figure 7b. Figure 7e shows the charge–discharge curve of the hybrid supercapacitor device at different specific currents (0.5 to 5 A g^−1^). The pseudo-symmetrical nature of the charge–discharge curve at all specific currents results in the good electrochemical performance of the hybrid device. The small IR drop at the start point of the discharge profile indicates the good conductivity of the fabricated device.

Appendix A shows the cycling stability obtained from the charge–discharge measurement for the Zn-Co-S//AC asymmetric hybrid supercapacitor device at a current density of 5 A g^−1^. Initially, the specific capacity decreased up to 30 cycles. Subsequently, the specific capacity increased and retained approximately 100% of its initial capacity after 2000 cycles. If further repeated redox reactions had been carried out, the capacity of the battery-type asymmetric hybrid device, Zn-Co-S//AC, could have been reduced. However, a symmetric EDLC-type device can show a very long cycling stability with the sacrifice of the energy density [36]. The specific energy and specific power of the Zn-Co-S//AC hybrid device were calculated using the following formulae [37,38]:(8)$$E=\frac{I\int Vdt}{m}$$
(9)$$P=\frac{E}{\Delta t}$$

Figure 7f presents the specific energy vs. specific power curve for the fabricated Zn-Co-S//AC asymmetric hybrid supercapacitor device obtained from the charge–discharge curves in Figure 7e. The Zn-Co-S//AC asymmetric hybrid device delivered a maximum specific energy of 19.0 Wh kg^−1^ at a specific power of 514.0 W kg^−1^ and a maximum specific power of 3.7 kW kg^−1^ at a specific energy of 4.96 Wh kg^−1^. This result reveals improved specific energy compared to other Zn-, Co-, and Zn-Co-based supercapacitor devices, such as Zn_x_Co_1-x_S//AC (14 Wh kg^−1^) [14], CoSx/C//PCNFs (15 Wh kg^−1^) [39], CoS_2_//AC (14 Wh kg^−1^) [40], and ZnMn_2_O_4_//AC (18 Wh kg^−1^) [41]. The vertically aligned nanostrips on the nickel foam provide a better diffusion pathway for electrolyte ions, a larger electrode surface area, and a faster electron transfer to the current collector. During device fabrication, only the strips grown on the top surface of the Ni foam can contact the separator and collapse. On the other hand, most of the strips (more than 95%) grown on the other sides and inside the porous structure remained intact.

## 4. Conclusions

Ni-foam-supported Zn-Co-S nanostrip arrays were synthesized using a facile two-step hydrothermal process. FE-SEM clearly showed that the Zn-Co-S nanostrip cluster arrays grew homogeneously on the skeleton of the 3D Ni foam. The Zn-Co-S nanostrip assembled together and formed nanostrip cluster arrays. These nanostrip cluster arrays provide better pathways for efficient ion/electron transport, enhancing the electrochemical activity. Zn-Co-S nanostrip cluster arrays exhibited a specific capacity of 830 C g^−1^, which is three times higher than the ZnCo_2_O_4_ electrode (300 C g^−1^) at a specific current of 2.0 A g^−1^. The high specific capacity of Zn-Co-S nanostrip arrays was attributed to the higher polarizability, lower electro-negativity, the larger size of the sulfur ion, and the shortest pathway of the nanostrip array. The hybrid device assembled using Zn-Co-S nanostrip cluster arrays as a positive electrode and AC as a negative electrode delivered a maximum specific energy of 19.0 Wh Kg^−1^ (at a specific power of 514 W kg^−1^) and a maximum specific power of 3.71 kW kg^−1^ (at a specific energy of 4.96 Wh kg^−1^). The binder-free Zn-Co-S nanostrip could be applied to various energy-storage applications and other electrochemistry-based devices.

## Figures and Tables

**Figure 1 nanomaterials-11-03209-f001:**
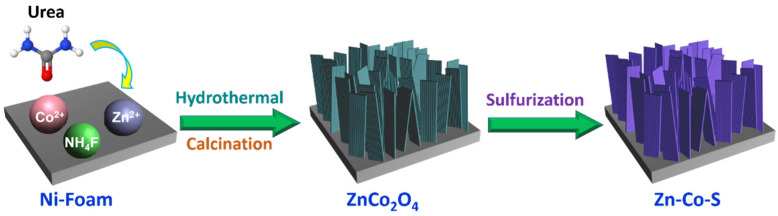
Schematic diagram for the preparation processes of the Zn-Co-S nanostrip cluster arrays.

**Figure 2 nanomaterials-11-03209-f002:**
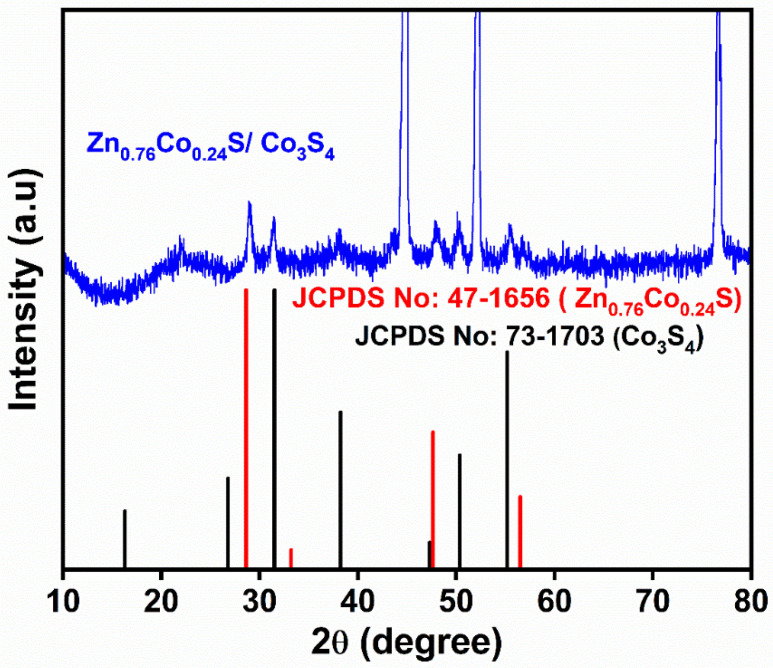
XRD patterns of the Zn-Co-S nanostrip cluster arrays.

**Figure 3 nanomaterials-11-03209-f003:**
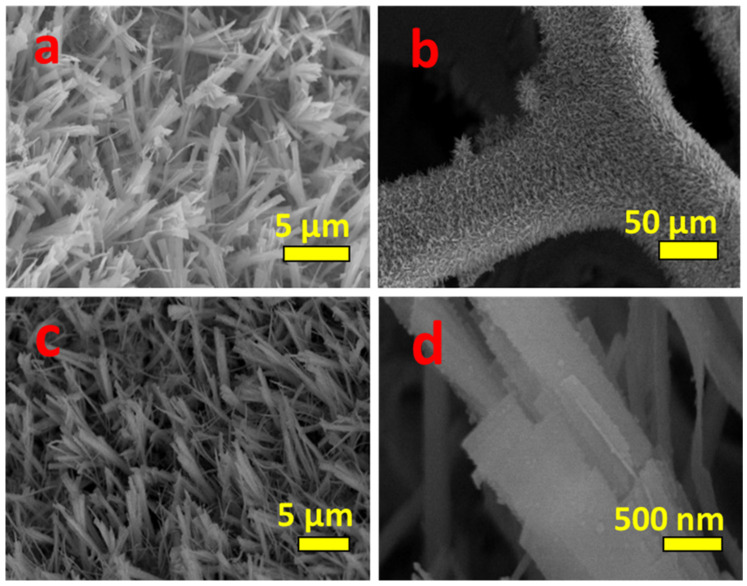
FE-SEM images of (**a**) ZnCo_2_O_4_ and (**b**–**d**) Zn-Co-S samples.

**Figure 4 nanomaterials-11-03209-f004:**
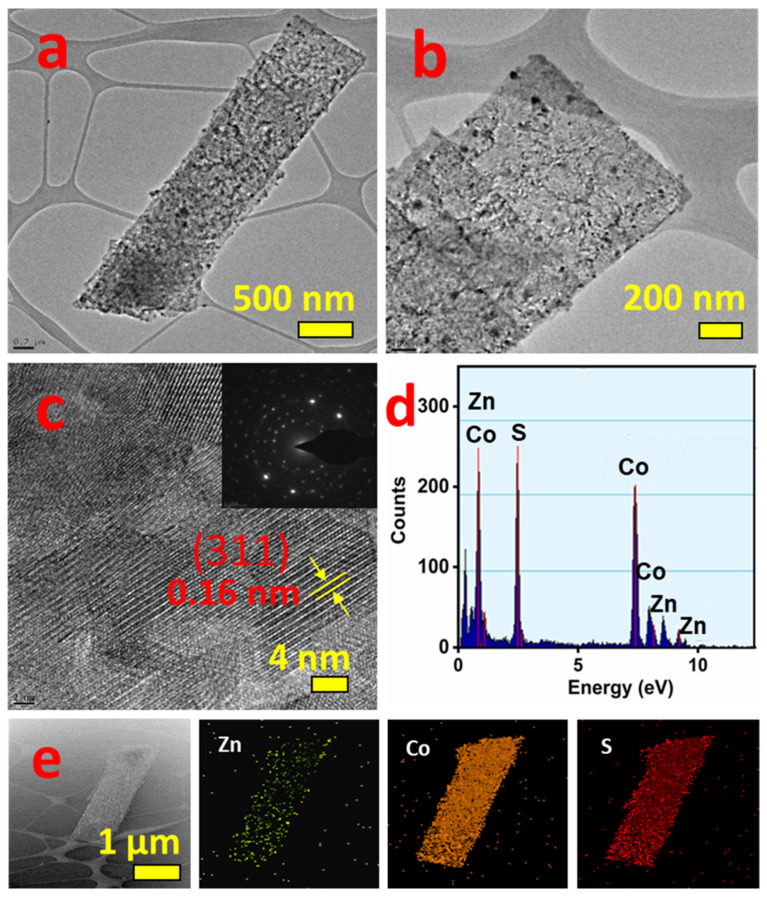
HR-TEM images (**a**–**c**), EDAX (**d**), and elemental mapping (**e**) of a Zn-Co-S nanostrip. The inset in (**c**) shows the SAED pattern.

**Figure 5 nanomaterials-11-03209-f005:**
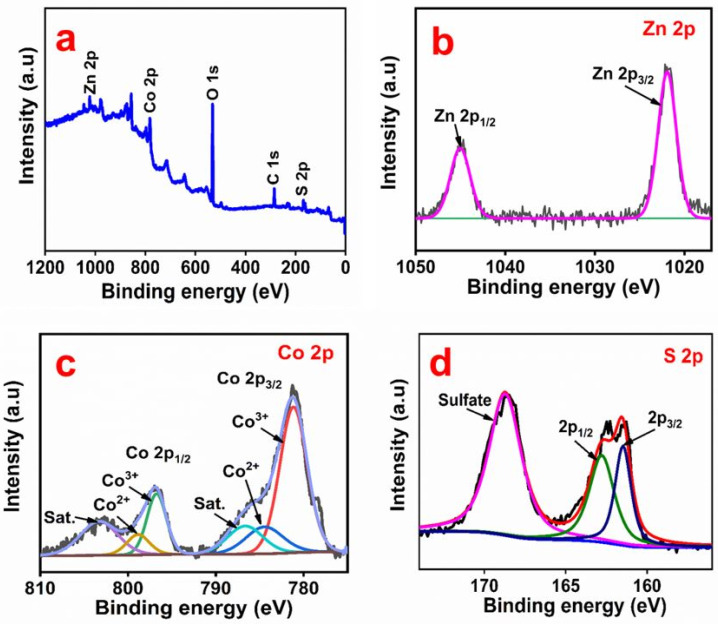
XPS survey spectrum (**a**) and high-resolution XPS spectra of Zn 2p (**b**), Co 2p (**c**), and S 2p (**d**) of the Zn-Co-S electrode material.

**Figure 6 nanomaterials-11-03209-f006:**
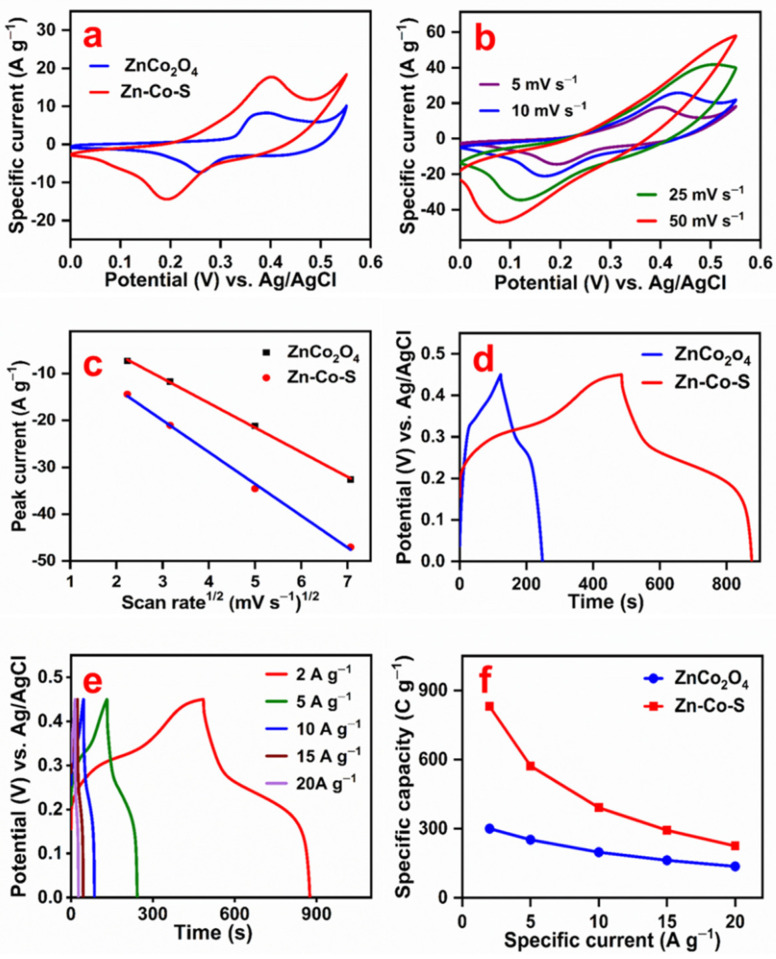
CV curves of the ZnCo_2_O_4_ and Zn-Co-S electrodes (**a**), CV curves of the Zn-Co-S electrode at different scan rates (**b**), the peak current vs. square root of scan rate (**c**), the charge–discharge curves of the ZnCo_2_O_4_ and Zn-Co-S electrodes (**d**), the charge–discharge curves of the Zn-Co-S electrode at different specific currents (**e**), and specific capacity vs. specific current of the ZnCo_2_O_4_ and Zn-Co-S electrodes (**f**).

**Figure 7 nanomaterials-11-03209-f007:**
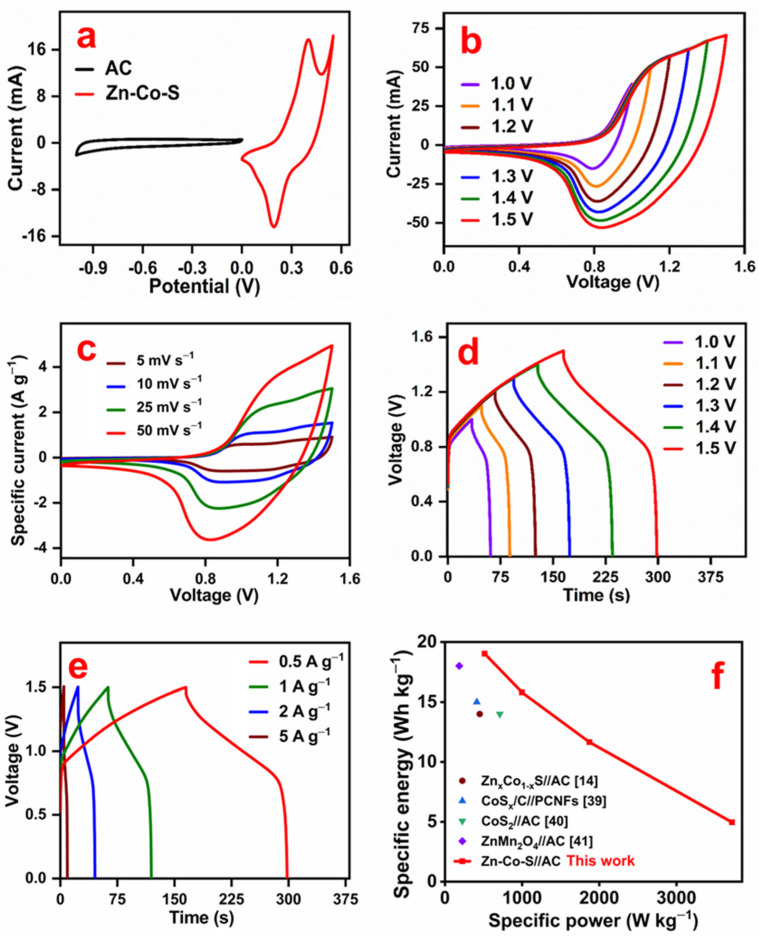
(**a**) CV curve of the AC and Zn-Co-S electrodes at 5 mV s^−1^, (**b**) CV curves of the Zn-Co-S//AC hybrid supercapacitor device at different voltages, (**c**) CV curves of the Zn-Co-S//AC hybrid supercapacitor device at different scan rates, (**d**) charge–discharge curves of the hybrid supercapacitor device at different voltages, (**e**) charge–discharge curves of the hybrid supercapacitor device at different specific currents, and (**f**) Ragone plots (specific energy vs. specific power) from the literature and this study.

## Data Availability

The data presented in this study are available in the lab research notebook in Yeungnam University.

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
