# Peer review of "Facile Synthesis of Zn-Co-S Nanostrip Cluster Arrays on Ni Foam for High-Performance Hybrid Supercapacitors"

_nanomaterials, 2021, doi:10.3390/nano11123209_

Round 1
Reviewer 1 Report
This manuscript presents synthesis of Zn-Co-S nanostrips on Ni-foam and construction of hybrid supercapcitors. Material characterization and electrochemical characterization results are given. The work in general is interesting and could be published but the following issues need to be addressed.
Figure S6 makes no sense. How can capacitance retention exceed 100%? That is physically impossible. What is your definition of this ratio anyway? For a common and normal looking behavior, see Figure 4 (b) in ACS Applied Energy Materials 2020, 3, 4965−4973, which shows only ~ 1% drop after 100,000 cycles for EDLC and 7% drop after 40,000 cycles for a hybrid supercapacitor.
Figures 1 and 3C seem to depict vertical structures. Is vertical orientation critical for performance? After all, when you press the two electrodes separated by a membrane separator, these vertical structures would get squished anyway.
Lines 119-121: “Ni-foam was used as a current collector for the preparation of Zn-Co-S because of its zig-zag skeleton and high porosity, which helps increase the active surface area.” Did you measure the porosity and surface area? How do you know these values are high?
Figure 7 has six panels but does not have much discussion about most panels. Each panel has one sentence about what the curves are like, which the reader can readily see. There should be more discussion, giving insight into what is going on.
One sentence comparing with previous literature at the end is not enough. The authors can give a Table in the Supplementary Information with multiple columns containing comparison of specific energy, specific power, specific capacitance, cyclic performance (if reported) etc.
Author Response
Thank you for giving me this opportunity to improve the manuscript. We have carried out the corrections based on the reviewer's comments. The point by point response for each reviewer is given as follows:
- Figure S6 makes no sense. How can capacitance retention exceed 100%? That is physically impossible. What is your definition of this ratio anyway? For a common and normal looking behavior, see Figure 4 (b) in ACS Applied Energy Materials 2020, 3, 4965−4973, which shows only ~ 1% drop after 100,000 cycles for EDLC and 7% drop after 40,000 cycles for a hybrid supercapacitor.
Answer: I agree with you. Good carbon materials can retain 99% of the initial capacitance even after 100,000 cycles. For hybrid capacitors, however, it can range from 95% to 80 %, depending on the positive electrode materials. Some people reported capacitance as low as 60% for some materials such as LDH.
We defined the capacitance retention as the ratio of the capacitance at a certain time to the initial capacitance. As you pointed out, the capacitance should not be over 100% with cycling if we have the best condition at the initial stage by just doing initial stabilization. In this research, unfortunately however, the first author skipped the initial stabilization, getting an increasing trend for capacitance as in many other papers. The electrode might have had air inside the pore, decreasing the diffusion of electrolyte and thus wettability of the electrode. Another reason could be lacking of full activation of the electrode material. Since this work was done four years ago and we do not have the same electrode, we cannot do the test again. I hope you can understand this situation.
- Figures 1 and 3C seem to depict vertical structures. Is vertical orientation critical for performance? After all, when you press the two electrodes separated by a membrane separator, these vertical structures would get squished anyway.
Answer: The vertically aligned nanostrips on the nickel foam provide a better diffusion pathway for electrolyte ions, a larger electrode surface area, and a faster electron transfer to the current collector. During the device fabrication, only the strips grown on the top surface of Ni foam can contact the separator and be collapsed. However, most of the strips (more than 95%) grown on the other sides and inside of the porous structure are all remained intact and therefore stay safe.
- Lines 119-121: “Ni-foam was used as a current collector for the preparation of Zn-Co-S because of its zig-zag skeleton and high porosity, which helps increase the active surface area.” Did you measure the porosity and surface area? How do you know these values are high?
Answer: Ni foam is a sheet (about 1.6 mm thick) but consists of a lot of large pores in the skeleton of a huge number of interconnecting bridges. The porosity is the ratio of the total void volume to the apparent volume of the entire structure. The surface area is the area of all surfaces of the bridge structures of nickel foam. We have been using a commercial nickel foam purchased by MTI (a USA company), which has a porosity of ≥ 95% (80-110 pores per inch and the average pore diameter about 0.25 mm). Due to this highly porous structure, the nickel foam can provide more surface for the growth of Zn-Co-S.
- Figure 7 has six panels but does not have much discussion about most panels. Each panel has one sentence about what the curves are like, which the reader can readily see. There should be more discussion, giving insight into what is going on.
Answer: Thank you for your suggestion. We have modified and added sentences (highlighted) to the discussions section of the manuscript.
“Figure 7a shows the CV curves of activated carbon and Zn-Co-S electrodes measured at 5 mV s−1 in a half-cell system. The rectangular-shaped CV curve of the AC electrode showed the electric double-layer capacitor behavior with a potential window from -1.0 to 0 V. The Zn-Co-S electrode showed a battery-type Faradaic redox peaks with a potential window from 0 to 0.55 V. CV measurements were carried out at different voltages (1.0 to 1.5 V) to get the optimal voltage window (Figure 7b). The curve at 1.55 V showed the small spike near the highest voltage, it is not included in this figure. As the curve at 1.5 V showed the largest area with distinct redox behavior, 1.5 V was chosen as the voltage window for the device. The CV curves of the asymmetric hybrid supercapacitor device (Zn-Co-S//AC) measured at different scan rates (Figure 7c) clearly show the characteristics of both the double layer capacitor and battery type electrode behavior as Figure 7b. For the increase in the scan rate from 5 to 50 mV s−1, no significant change or shift of the redox peaks was observed, which indicates a good charge delivery rate of the fabricated device.
Figure 7d presents the charge-discharge curve of the asymmetric hybrid supercapacitor device at different voltages, ranging from 1.0 to 1.5 V, which is consistent with the CV curves in Figure 7b. Figure 7e shows the charge-discharge curve of the hybrid supercapacitor device at different specific currents (0.5 to 5 A g‒1). The pseudo-symmetrical nature of the charge-discharge curve at all specific currents results in the good electrochemical performance of the hybrid device. The small IR drop at the start point of the discharge profile indicates a good conductivity of the fabricated device. Figure 7f displays a specific energy vs. specific power curve for the fabricated Zn-Co-S//AC asymmetric hybrid supercapacitor device obtained from the charge-discharge curves in Figure 7e. The cycling stability were obtained from the charge-discharge measurement at a current density of 5 A g‒1 (Figure S6). The Zn-Co-S//AC asymmetric hybrid device delivered a maximum specific energy of 19.0 Wh kg−1 at a specific power of 514.0 W kg−1 as well as a maximum specific power of 3.7 kW kg−1 at a specific energy of 4.96 Wh kg−1. This result reveals improved specific energy as compared with other Zn-, Co-, and Zn-Co-based supercapacitor devices such as ZnxCo1-xS//AC (14 Wh kg−1) [13], CoSx/C//PCNFs (15 Wh kg−1) [37], CoS2//AC (14 Wh kg−1) [38], and ZnMn2O4//AC (18 Wh kg−1) [39]. The vertically aligned nanostrips on the nickel foam provide a better diffusion pathway for electrolyte ions, a larger electrode surface area, and a faster electron transfer to the current collector. During the device fabrication, only the strips grown on the top surface of Ni foam can contact the separator and be collapsed. However, most of the strips (more than 95%) grown on the other sides and inside of the porous structure are all remained intact and therefore stay safe.”
- One sentence comparing with previous literature at the end is not enough. The authors can give a Table in the Supplementary Information with multiple columns containing comparison of specific energy, specific power, specific capacitance, cyclic performance (if reported) etc.
Answer: We have included a comparison table in the revised Supporting Information file (Table S1).

Reviewer 2 Report
The manuscript reports that Zn-Co-S nanostrip cluster arrays as supercapacitor electrodes show high electrochemical performance, which can attract the readers' interests. However, there are some issues should be corrected before its acceptance/publication.
1) The authors claimed that "The higher specific capacity of the Zn-Co-S electrode than the ZnCo2O4 electrode may be due to the higher conductivity and synergic effect ..." . "May be..." should not appear in paper. The authors should provide direct and convincing proofs.
2) Nitrogen adsorption/desorption isotherms of the electrode materials should be provided.
3) The abstract, conclusions and introduction had better be refined to highlight the novelty and significance of this study.
Author Response
Thank you for giving me this opportunity to improve the manuscript. We have carried out the corrections based on the reviewer's comments. The point by point response for each reviewer is given as follows:
- The authors claimed that "The higher specific capacity of the Zn-Co-S electrode than the ZnCo2O4 electrode may be due to the higher conductivity and synergic effect ..." . "May be..." should not appear in paper. The authors should provide direct and convincing proofs.
Answer: Thank you for pointing out this issue. We removed the term “may be..” in the manuscript and provided convincing evidence for higher specific capacity of the Zn-Co-S electrode than the ZnCo2O4 electrode: (i) zinc cobalt sulfide has a lower band gap and much higher electrical conductivity than zinc sulfide, cobalt sulfide, and zinc cobalt oxide and (ii) the binary-metal sulfide Zn-Co-S provides a higher redox potential than single metal sulfides such as ZnS, CoS, CoS2, and Co3S4. These two can be the reasons for better electrochemical performance. A synergistic effect of the two different metal ions of the Zn-Co-S electrode must have enhanced the performance.
- Nitrogen adsorption/desorption isotherms of the electrode materials should be provided.
Answer: You are right. We should have measured the BET surface area. To measure an accurate BET data, we must have a sample of more than 300 mg. However, our electrode material was directly grown on nickel foam. To collect the sample, we need to scrape the strips from the nickel foam, which has majority surface is inside the foamy structure. The collecting 300 mg by scraping from the pores of nickel foam is very difficult. Since this work was done four years ago, we do not have the sample anymore. Also, the first author returned to his home country 3.5 years ago. Therefore, making a new electrode material is very difficult at this stage. Please understand our situation.
- The abstract, conclusions and introduction had better be refined to highlight the novelty and significance of this study.
Answer: Thank you suggestion. Based on your comment, we have modified the abstract, conclusion, and introduction of the manuscript.

Reviewer 3 Report
Manuscript entitled “Facile synthesis of Zn-Co-S nanostrip cluster arrays on Ni-foam for high-performance hybrid supercapacitors” has been reviewed. In this work, Zn-Co-S nanoribbon cluster arrays were grown in situ on nickel foam by a two-step hydrothermal method and used as electrode materials for supercapacitors, showing relatively good electrochemical performance which is interesting. However,many issues need to be addressed before the manuscript can be accepted in Nanomaterials.
1. A relatively detailed analysis and review of the research work is very necessary. The authors reported the Zn-Co-S nanoribbon cluster arrays grown in situ on nickel foam. Why did the authors choose Zn and Co as the source of transition metals involved in the Faraday reaction? The author did not elaborate on this point. As the author mentioned, few people have studied it, and this direction is correct and encouraged. However, the explanation that "there is relatively little research" is still unsatisfactory. The author should give some specific reasons, such as the electronic structure, the synergistic effect of Zn/Co bimetal and so on.
2. Transition metal-based heteroatom compounds, such as various N doping, S doping and P doping, have been extensively studied and proved to be a feasible strategy for electrochemical energy storage. And the writing of the paper should be very logical, especially in a scientific paper, the Introduction part of the research work to carry out a logical and relevant elaboration is absolutely important. Therefore, as a double transition metal sulfide, it is suggested that the author should explain in a progressive way the reasons for the high electrochemical storage performance of the target material in the Introduction. For example, it should explain why the performance of the material obtained by the first hydrothermal method is not good, and what are its limitations, and then the subsequent S element doping can be derived. In addition, some published related work should be referred to, such as (ELECTROCHIM ACTA 2021, 138200)、(ELECTROCHIM ACTA 2021, 138433).
3. What is the mass of the active material per unit area for the electrode material grown in situ? How to calculate it? If it is set as the active material through the difference in the quality of the reaction before and after, will the nickel foam participate in the reaction during the synthesis process and change the quality of the current collector (nickel foam)?
4. As a supercapacitor electrode material, one of its advantages over other electrochemical energy storage devices is its excellent cycle stability. Therefore, the cycle test with 2000 cycles seems to be not enough. If possible, it would be better to increase the number of cycles to 5000.
5. There are many grammatical and format errors throughout the paper. Some sentences cannot be read smoothly, there are no spaces between words, and there are no spaces between units and values. The author should carefully check and correct the language.
Author Response
Thank you for giving me this opportunity to improve the manuscript. We have carried out the corrections based on the reviewer's comments. The point by point response for each reviewer is given as follows:
- A relatively detailed analysis and review of the research work is very necessary. The authors reported the Zn-Co-S nanoribbon cluster arrays grown in situ on nickel foam. Why did the authors choose Zn and Co as the source of transition metals involved in the Faraday reaction? The author did not elaborate on this point. As the author mentioned, few people have studied it, and this direction is correct and encouraged. However, the explanation that "there is relatively little research" is still unsatisfactory. The author should give some specific reasons, such as the electronic structure, the synergistic effect of Zn/Co bimetal and so on.
Answer: Thank you for the comments. Following your suggestion, we have added the reason why we chose Zn and Co and the previous literature in this area in the introduction section.
“Cobalt sulfides provide the highest capacitances in aqueous electrolytes among all the transition metal oxides and sulfides. However, Co is an expensive material. Many research has been done to replace Co with less expensive metals such as Ni, Zn, and Cu. Among them, zinc is the most abundant, the least expensive, and relatively nontoxic. Therefore, Zn was chosen to reduce the use of Co, expecting a synergistic effect of Zn with larger atomic size than Co.”
- Transition metal-based heteroatom compounds, such as various N doping, S doping and P doping, have been extensively studied and proved to be a feasible strategy for electrochemical energy storage. And the writing of the paper should be very logical, especially in a scientific paper, the Introduction part of the research work to carry out a logical and relevant elaboration is absolutely important. Therefore, as a double transition metal sulfide, it is suggested that the author should explain in a progressive way the reasons for the high electrochemical storage performance of the target material in the Introduction. For example, it should explain why the performance of the material obtained by the first hydrothermal method is not good, and what are its limitations, and then the subsequent S element doping can be derived. In addition, some published related work should be referred to, such as (ELECTROCHIM ACTA 2021, 138200)、(ELECTROCHIM ACTA 2021, 138433).
Answer: Thank you for your comments. We have modified the manuscript (Introduction) accordingly and cited the suggested article.
“Due to the higher polarizability, lower electro-negativity, and larger size of the S2− ion, the substitution of oxygen with sulfur increases the conductivity of the electrode and ion diffusivity [10].”
- What is the mass of the active material per unit area for the electrode material grown in situ? How to calculate it? If it is set as the active material through the difference in the quality of the reaction before and after, will the nickel foam participate in the reaction during the synthesis process and change the quality of the current collector (nickel foam)?
Answer: We have modified the discussion section (3.1) according to your comment.
“The mass loading of the active material on a Ni foam was approximately 2.5 mg cm-2, which was calculated from the mass difference before and after the synthesis reaction. Because we cleaned the Ni foam with HCl, the possibility of reaction of the cleaned Ni foam with metal-oxide precursors should be low, resulting in no considerable change in the quality of Ni foam.”
- As a supercapacitor electrode material, one of its advantages over other electrochemical energy storage devices is its excellent cycle stability. Therefore, the cycle test with 2000 cycles seems to be not enough. If possible, it would be better to increase the number of cycles to 5000.
Answer: Yes, you are right. We should have done at least 10,000 cycles. Four to five years ago when this work was done, cycling tests of 2,000-3,000 cycles were accepted. Unfortunately, the first author returned to his country 3.5 years ago. Therefore, we cannot do the further cycling test. Please understand our situation.
- There are many grammatical and format errors throughout the paper. Some sentences cannot be read smoothly, there are no spaces between words, and there are no spaces between units and values. The author should carefully check and correct the language.
Answer: Thank you for your comment. The manuscript was corrected by a professional native English-speaker proofreader but his English correction was not complete. Now, we asked him to check the entire manuscript again. Now, I think he removed the flaws and modified them to an acceptable form.

Round 2
Reviewer 1 Report
The authors have done moderate revision. I can accept their excuse regarding my first question for which they responded by saying that the samples are 4 years old. They also expanded their discussion which is good.
I am surprised that they did not cite the reference I suggested. That must be cited so the readers know 100000 cycles are possible, as most papers stop at few thousand cycles; the readers will also know that it is possible to lose only a fraction of % after that many cycles.
Author Response
- The authors have done moderate revision. I can accept their excuse regarding my first question for which they responded by saying that the samples are 4 years old. They also expanded their discussion which is good.
I am surprised that they did not cite the reference I suggested. That must be cited so the readers know 100000 cycles are possible, as most papers stop at few thousand cycles; the readers will also know that it is possible to lose only a fraction of % after that many cycles.
Answer: Thank you. In the reviewer’s suggestion, I was not advised to refer to the reference. The reviewer’s suggestion was right and we answered he is right. However we cannot change it as it was done four years ago. A reviewer must deliver a clear message to the authors what to do. These days tests of 10,000 cycles are standard. When the research was done the standard was 3000-5000 cycles. Wouldn’t it be strange to the readers if we say 100,000 cycles are possible while we did only 2000 cycles?
Now, we have included the paper based on the reviewer’s suggestion and modified the text as shown below:
“If further repeated redox reactions had been carried out, the capacity of the battery-type asymmetric hybrid device, Zn-Co-S//AC, could have been reduced. However, a symmetric EDLC-type device can show a very long cycling stability at the sacrifice of the energy density [36].”
